# Programming crack patterns with light in colloidal plasmonic films

Fanny Thorimbert[1], Mateusz Odziomek[2], Denis Chateau[3], Stéphane Parola [3] & Marco Faustini [1,4] ✉

Crack formation observed across diverse fields like geology, nanotechnology, arts, structural engineering or surface science, is a chaotic and undesirable phenomenon, resulting in random patterns of cracks generally leading to material failure. Limiting the formation of cracks or "programming" the path of cracks is a great technological challenge since it holds promise to enhance material durability or even to develop low cost patterning methods. Drawing inspiration from negative phototropism in plants, we demonstrate the capability to organize, guide, replicate, or arrest crack propagation in colloidal films through remote light manipulation. The key consists in using plasmonic photothermal absorbers to generate "virtual" defects enabling controlled deviation of cracks. We engineer a dip-coating process coupled with selective light irradiation enabling simultaneous deposition and light-directed crack patterning. This approach represents a rare example of a robust self-assembly process with long-range order that can be programmed in both space and time.

Crack propagation occurs commonly in nature, encompassing various phenomena from geological processes[1,2] to living organisms such as the skin of crocodiles[3] and elephants[4]. In synthetic materials, crack formation is typically an uncontrolled and chaotic process leading to the formation of random patterns of cracks which is associated with material failure[5–7]. Limiting formation of cracks or controlling their propagation and the geometry of the crack patterns is one of the greatest technological challenges in a number of domains. Additionally, an emerging trend in the field seeks to leverage this issue and turn it into an advantage toward a patterning method at high-resolution and high-throughput patterns[8]. For instance, crack formation can be controlled by utilizing preexisting defects to guide crack propagation in SiN films[9] or polymeric materials[10,11]. This enables the fabrication of intricate patterns for diverse applications, including flexible electronics[12,13], micro- and nanofluidics[14], optical devices[15], and sensors[16,17]. However, these crack-directed processes require the prefabrication of defects. In contrast, under specific conditions, the spontaneous ordering of cracks can be observed in evaporating colloidal solutions:[18–21] cracks propagate perpendicular to the evaporation front forming arrays of linear cracks with uniform spacing, without the need for preformed defects[22]. This method can be applied to polymers[23–25], sol-gel oxides[18,26,27], Metal-Organic Frameworks films[28], and the transfer of patterns to other materials such as gold[29] and perovskite[30]. However, crack patterns in colloidal films are typically limited to random or linear cracks. Extending this approach to complex curved patterns geometries, where both ordering and orientation can be controlled remains a challenge. More ambitiously, by utilizing light to program directional growth of cracks or control the self-ordering process would enable the emergence of unprecedented life-like responsiveness reminiscing phototropism in living plants.

In this study, we prove that self-assembly and propagation of cracks in drying polymer colloidal films can be manipulated by light by local thermoplasmonic effects. We engineer various colloidal plasmonic systems to spatially and temporally program the heating zones,

[1]Sorbonne Université, CNRS, UMR 7574, Chimie de la Matière Condensée de Paris, F-75005 Paris, France. [2]Colloid Chemistry Department, Max Planck Institute of Colloids and Interfaces, Am Mühlenberg 1, 14476 Potsdam, Germany. [3]Ecole Normale Supérieure de Lyon, CNRS UMR 5182, Université Claude Bernard Lyon 1, Laboratoire de Chimie, 46 allée d'Italie, F69364 Lyon, France. [4]Institut Universitaire de France, Paris, France. ✉e-mail: marco.faustini@sorbonne-universite.fr

acting as "virtual defects". More specifically, we investigate two systems: gold dewetted nanoparticles and gold bipyramids (Au BPs), which convert light into heat in the visible or near-infrared range, respectively. Moreover, we chose these two plasmonic systems to explore two different strategies: integrating the photothermal heaters on the substrate or within the colloidal solution. Drawing an analogy to negative phototropism in plant roots[31], we demonstrate that propagating cracks can "escape" from light, self-replicate, or arrest depending on how the light is applied.

## Results

### Negative phototropism of cracks in a drying colloidal droplet

The general concept is depicted in the simple experiment shown in Fig. 1. The experiment involves drying a droplet of an aqueous colloidal solution containing 65 nm-polystyrene (PS) nanoparticles functionalized with Pluronic F-127, as confirmed by infrared spectroscopy (Supplementary Fig. 1, 2). In general, drying colloidal droplet results in a colloidal film at the edge due to the coffee ring effect[32]. Evaporation-induced internal stresses provoke the nucleation and propagation of cracks in the colloidal films that typically results random mud-like patterns. Instead, in our specific case, crack self-ordering occurs in the

form of aligned evenly-spaced cracks (Supplementary Fig. 3). This self-ordering process is illustrated in Fig. 1a. Three distinct zones can be distinguished: the colloidal solution, the colloidal gel filled with water and the dried film, as also displayed in the optical micrograph, in Supplementary Fig. 4[25]. The three zones are separated by the gelification front and the drying front. Cracks appear in the gel state, at a distance that is indicated as cracking front. The crack spacing results from the maximum global stress resulting from stress relaxation (due to crack opening) and stress accumulation (due to evaporation through the newly formed crack)[18]. Importantly, these periodic cracks always propagate perpendicularly to the meniscus. Consequently, circular droplets give rise to radial cracks propagating from the edge towards the center of the droplet (Supplementary Fig. 3).

In our experiment, the path of the propagating cracks can be manipulated by light, as illustrated in Fig. 1b, c. To do so, we dried the colloidal droplet on a glass substrate coated with plasmonic gold nanoparticles obtained through thermal dewetting (Supplementary Fig. 5a). The absorption peak of gold plasmonic substrate shows maximum of around 560 nm (Supplementary Fig. 5b). The surface was exposed to a 532 nm laser from a dark-field condenser positioned underneath (Fig. 1b) and the edge of the droplet (Fig. 1c). The video

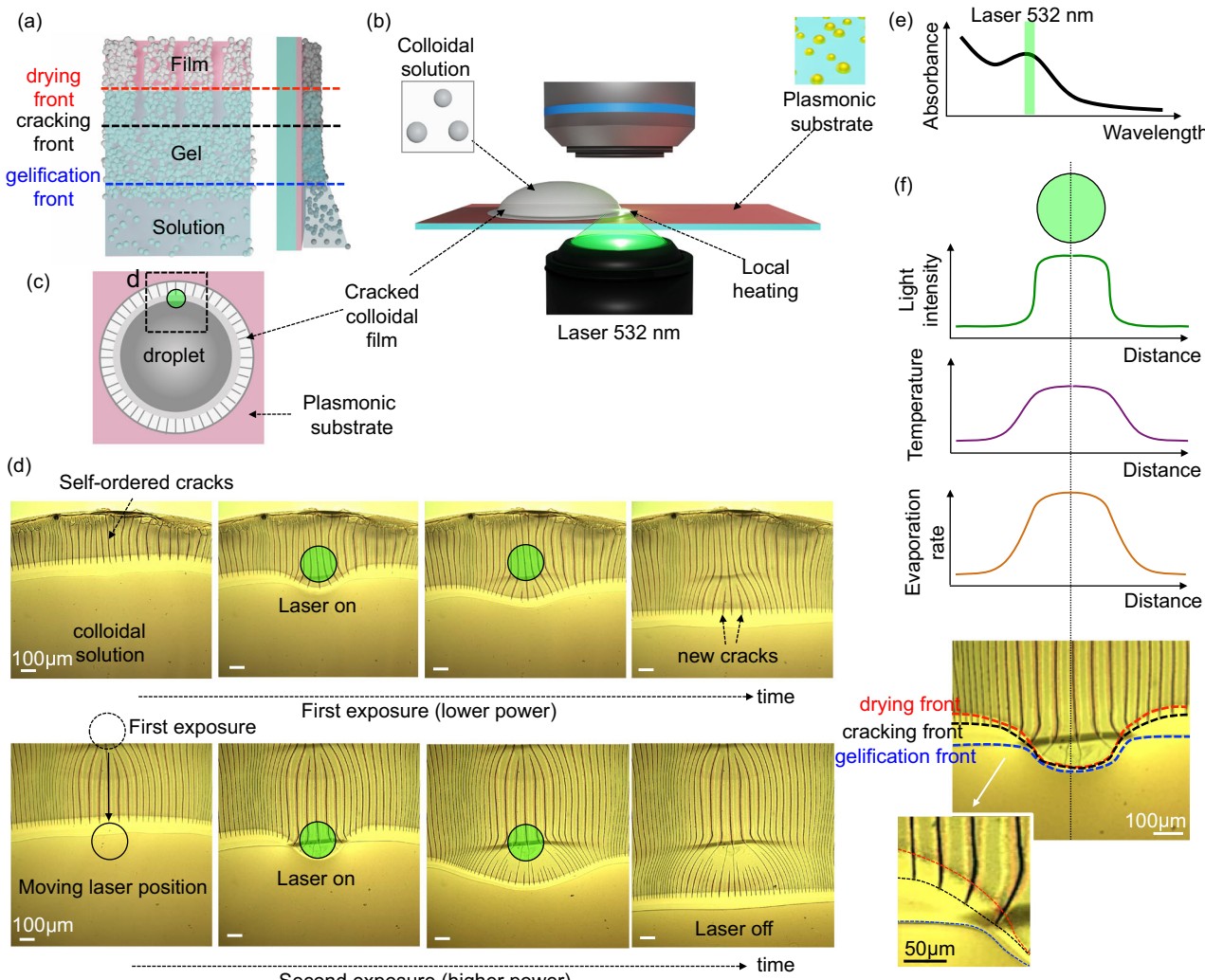

**Fig. 1 | Light-deviated crack propagation in a drying droplet. a** Scheme of the formation of periodic cracks during drying of a colloidal droplet (**b**) Scheme of the experimental set-up, a drying colloidal droplet on a plasmonic substrate, illuminated by a condensed 532 nm laser and **c** top-view of the drying colloidal droplet. **d** Optical microscopy image sequence of the crack propagation when a laser beam is applied: first and second lines with the laser's power set at 0.19 and 0.67 W/cm² respectively. **e** Relationship between plasmonic absorbance and laser wavelength to obtain high photothermal heating. **f** Qualitative evolution profiles of the light intensity, the temperature and the evaporation rate as function of the distance. Optical micrograph of the deformed meniscus (inset: zoom of the deviated part).

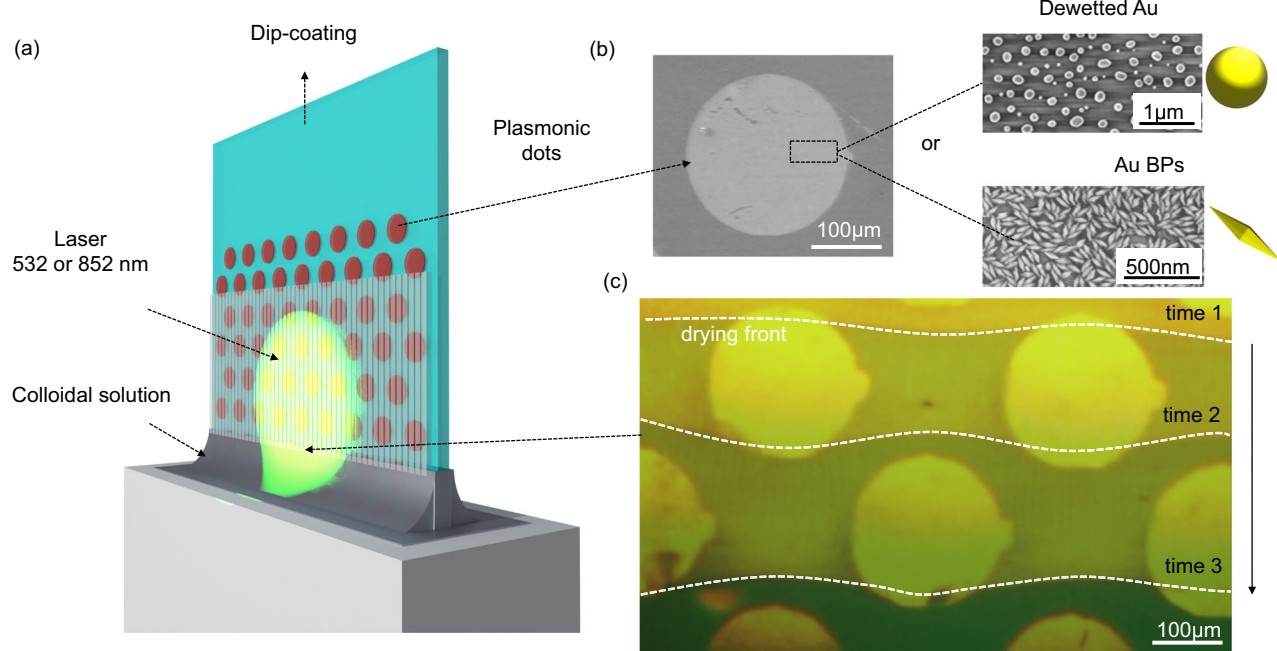

**Fig. 2 | Light-directed self-ordering of cracks coupled with dip-coating on a patterned plasmonic substrate. a** Scheme of the experimental set-up. **b** SEM micrographs of a plasmonic dot composed of either Au dewetted nanoparticles or bipyramids on glass. The small symbols will be used as a graphical guide to distinguish the samples made of Au dewetted nanoparticles (yellow spheres) or bipyramids (yellow bipyramids). **c** Optical micrograph illustrating the temporal evolution of the drying line during dip-coating and exposition to the laser beam.

frames presented in Fig. 1d (Supplementary Movie 1, 2) illustrate the evolution of the cracks formed at the edge of the droplet as illustrated in Fig. 1c during the drying, as observed under an optical microscope.

Prior to exposure, the colloidal film consists of evenly spaced self-organized cracks that propagate perpendicular to the meniscus. Upon turning on the laser for the first exposure, local irradiation results in a deformation of the gelification, cracking and drying fronts and the formation of a concave profile. After exposure, the fronts return linear, and the cracks continue to propagate in a straight direction. Similarly, when the sample is irradiated again with a higher-power laser (Fig. 1d, second exposure), a greater deviation of the cracks is observed. Additionally, when the spacing between adjacent cracks becomes too large, one or more new cracks nucleate in the middle space. This light-driven "self-replication" of cracks can be seen in Supplementary Fig. 6, which illustrates the final cracked pattern from Fig. 1d and displays three "generations" of cracks: those present at the start of the experiment and those formed after the first or second light exposures (this aspect will be discussed later). It is important to note, that when no plasmonic gold nanoparticles are present on the substrate, no deviation occurs (Supplementary Fig. 7, and Supplementary Movie 3).

We attributed the origin of the light directed self-ordering to evaporation driven photo-thermal effect. To achieve high efficient photo-heating, the laser wavelength ideally needs to match the absorbing specific wavelengths on the plasmonic nanoparticles as illustrated in Fig. 1e[33]. When the nanoparticles are exposed to laser irradiation at their resonance frequency, a maximun of energy from the absorbed photons is converted into heat. Irradiation at different resonances results in lower photothermal heating. To provide a general understanding of how photothermal heating is used to deviate cracks, in Fig. 1f, we qualitatively plotted the expected profiles of the light intensity, the temperature and the evaporation rate as function of the distance. Local laser irradiation results in the formation of a hot spot due to photo-thermal heating. According to Hertz-Knudsen equation[34], this leads to higher saturated vapor pressure and evaporation rate in the exposed zone. This results in (i) a higher loss of

water, responsible for the concave profile and (ii) a decrease of the distance between the gelification and drying fronts as reported previously[35]. Figure 1f shows that cracks always propagate perpendicularly to the cracking front and they are "deviated away" from the light beam. The concept of "negative phototropism" is further demonstrated in Supplementary Movie 4 and Supplementary Fig. 8, where a visible green laser beam is manually moved during drying to deviate the cracks away from the laser beam and create intricate crack paths.

## Light-directed self-ordering of cracks on plasmonic arrays

To expand the scope beyond circular droplets, we employed dip-coating, a widely used process for fabricating films from solutions[34,36], to engineer the light-directed self-ordering of cracks. Micrometric-thick colloidal films were made at a low withdrawal speed (0.004 mm/s) in the capillary/evaporative regime[37,38]. As the solution underwent directional drying, arrays of parallel and periodically ordered cracks were formed (Fig. 2a). To induce localized photo-thermal heating, the glass substrate was patterned with arrays of micrometric circular dots, made by photo-lithography, composed of plasmonic gold nanoparticles. We investigated two systems: either dewetted nanoparticles (as obtained by dewetting as described above) or bipyramids (Au BPs), applied by drop casting as depicted in the SEM micrograph of Fig. 2b. In the following, we mainly focus on the circular plasmonic dots composed of Au dewetted nanoparticles. As illustrated in Fig. 2a, during dip-coating the sample was irradiated by a 532 nm laser (beam size 4.52 mm², 6.75 W/cm²). In this configuration, each plasmonic dots act as a hot spot with a temperature of around 50 °C estimated by an IR camera (Supplementary Fig. 9). The crack formation process during irradiation and dip-coating was observed using a custom optical setup, as shown in Supplementary Movie 5. Figure 2c presents an optical micrograph in which we displayed the evolution of the drying front at different times in the zone exposed by the laser. As discussed earlier, the local heating on the plasmonic dots induces a deformation of the cracking front, resulting in a concave profile (Supplementary Fig. 10). In the case of the hexagonal

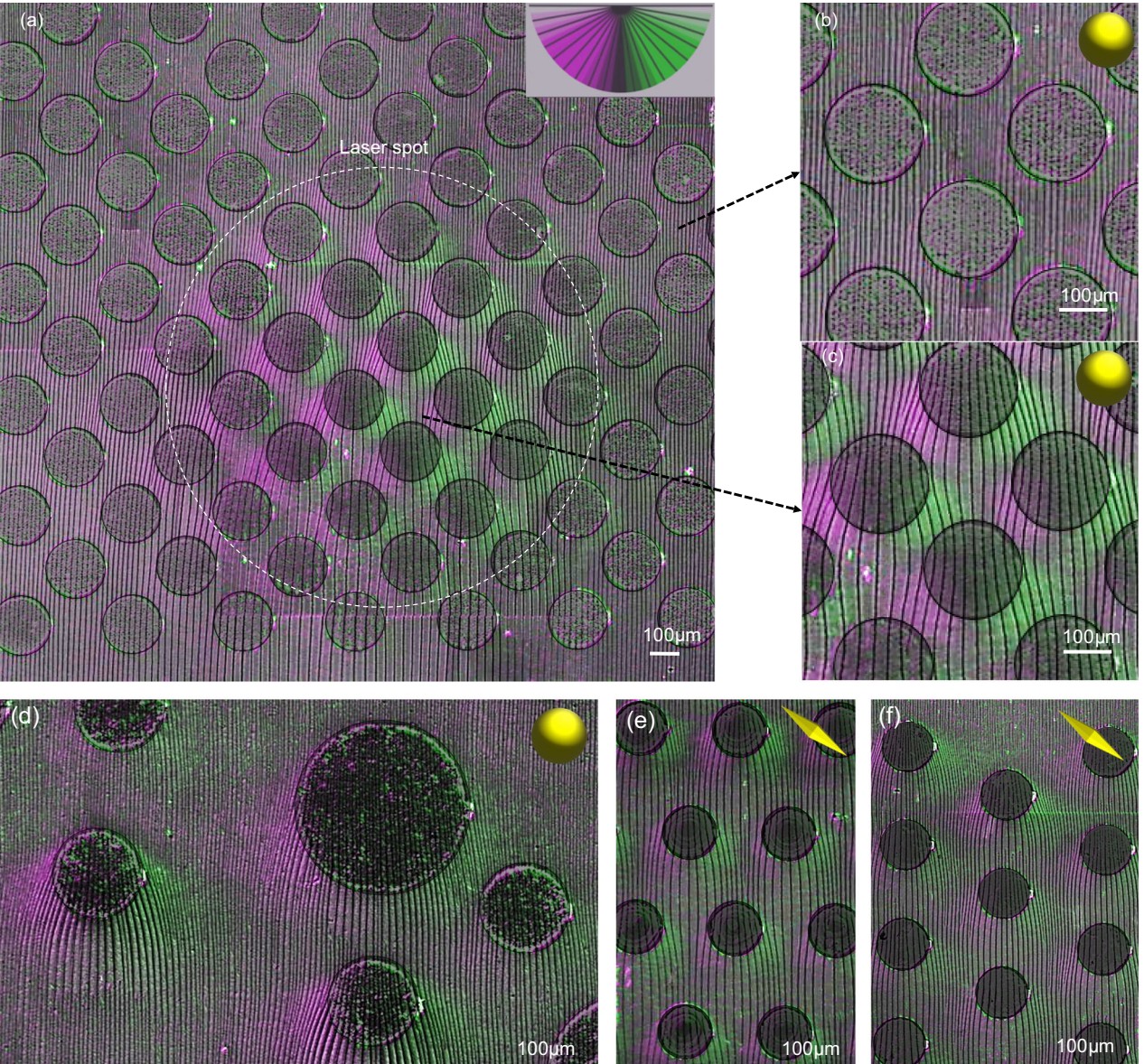

**Fig. 3 | Examples of cracked patterns obtained on plasmonic substrates.**
**a** Micrographs of a hexagonal pattern with deviated cracks, represented in pink when it is deviated on the left, and green when it is deviated on the right.
**b**, **c** Zoomed micrographs of the crack pattern in a not irradiated and irradiated area respectively. **d** Optical microscopy image of a cracked pattern on a random plasmonic array. **e**, **f** Optical microscopy images of two different hexagonal patterns of plasmonic dots made of gold bipyramid substrate, and illuminated by a 852 nm laser.

array of plasmonic dots in Fig. 2c, this process is observed for each dot: the drying front exhibits a continuously changing wavy profile while dip-coating. The final crack pattern is depicted in the micrograph of Fig. 3a. The dashed line represents the illuminated zone, while the darker circular spots represent the plasmonic dots. The deviation of cracks is illustrated by a colored map, in which crack orientation and angle have assigned colors (purple when turning left or green when turning right). Outside of the irradiated area, the cracks propagate straight (Fig. 3b) indicating that the presence of the plasmonic dots alone does not affect the self-ordering process. Conversely, in the irradiated zone, cracks (Fig. 3c) form a regular wavy pattern in agreement with the evolution of the drying line shown in Fig. 2c.

The crack patterns can be tuned by simply changing the arrangement and size of plasmonic dots (Fig. 3d). We also performed the same experiment with colloidal AuBPs, stabilized with poly-ethylene glycol (PEG). These materials are excellent materials for photo-thermal heating and exhibit a sharp plasmonic peak around

850 nm in water confirming the high homogeneity in particle size and shape[39] (Supplementary Fig. 11). Plasmonic dots were prepared by drop-casting a colloidal solution onto patterned photoresists. While the distribution of AuBPs is not homogeneous and the particles are partially aggregated (Supplementary Fig. 12b), similar wavy cracked patterns can be observed when irradiating the surface with a 852 nm laser (beam size 28 mm$^2$ and power 3.4 W/cm$^2$) during dip-coating, Fig. 3e, f; while not optimized, these experiments further confirm our approach.

## Light-deviated cracks in plasmonic colloidal films
The previous examples typically require pre-patterned plasmonic substrates to manipulate cracks with light. To overcome this limitation, we generalize the concept of "negative phototropism" of cracks to any arbitrary substrate. Instead of adding a heating element to the substrate, the idea is to integrate the absorbing plasmonic AuBP ($\lambda_{SP} = 840$ nm) directly into the PS colloidal solution. Both colloidal

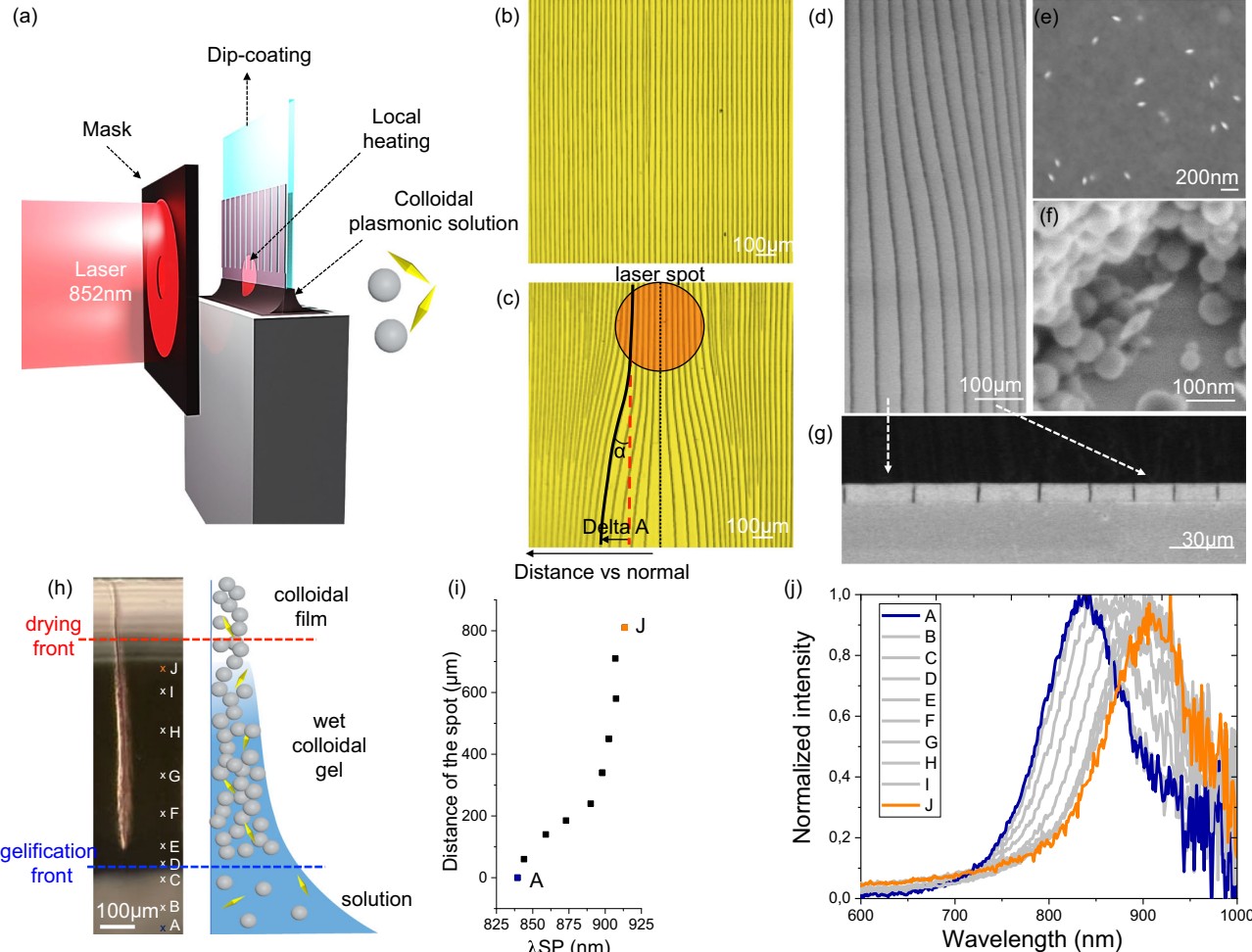

**Fig. 4 | Light-directed crack propagation obtained by dip-coating with a colloidal plasmonic solution. a** Scheme of the experimental dip-coating set-up of the formation of self-ordered cracks, light-deviated by a 852 nm laser. The localized heating of the colloidal plasmonic solution is obtained by irradiating the surface through a mask, placed in between the laser and the dip-coater. The grey spheres represent PS particles, whereas the yellow bipyramids represent the AuBP. **b** Optical microscopy image of non-deviated cracks. **c** Optical microscopy image of deviated cracks. The red circle represents the spot of light. Each crack starting from the normal is deviated with an angle equal to α and a distance equal to Delta A. Top-view SEM micrographs of a **d** deviated section of a cracked AuBP/PS film, with **e, f** high magnification images of the AuBPs and PS particles distribution. **g** SEM micrograph displaying a cross-section view of a deviated cracked zone. Hyperspectral analysis of a single crack formation during the drying of a colloidal plasmonic droplet results in an image where each pixel provides a scattering spectrum, enabling spatial mapping of the plasmonic scattering. **h** Dark-field optical microscopy image of a drying droplet of colloidal plasmonic solution with the formation of cracks and the corresponding scheme representing the concentration in nanoparticles from the solution to the formation of a wet colloidal gel. **i** Graphical plot of the distance of the spot, from A to J, as function of the maximum of the plasmon resonance peak. **j** Scattering spectra of the evaporating colloidal plasmonic solution depending on the distance. Source data of Figure **i, j** are provided as a Source Data file.

materials were mixed in a concentration ratio AuBP/PS of $5.21 * 10^{-3}$. Supplementary Fig. 13 illustrates that the local temperature of the AuBP/PS solution can be increased up to 100 °C as function of the power of the 852 nm laser.

Figure 4a presents the approach to induce a single localized hot spot during dip-coating. By projecting the laser through a single-hole mask (place in front of the sample), a single hot spot is created (laser power of 3.4 W/cm²). Without applying any light, parallel periodic cracks are obtained by dip-coating, as shown in Fig. 4b. Instead, in presence of light a divergent crack pattern is observed in the illuminated area (red circle) Fig. 4c.

The morphology of the AuBP/PS colloidal film was investigated using Scanning Electron Microscopy (SEM). Figure 4d, e, f shows the SEM micrograph of a deviated section indicating that the PS and AuBP colloids were homogenously distributed, with brighter spots representing the AuBP and further confirmed in Supplementary Fig. 14. In addition, SEM analysis in Fig. 4g confirm that a similar thickness is observed in large and low spacing regions, ruling out possible major

modifications in the deposition process. We further studied the evolution of the optical properties of the plasmonic colloidal system during crack formation using in situ dark field hyperspectral microscopy in reflection mode (Supplementary Fig. 15). The obtained hyperspectral image provides the scattering spectra of each pixel, enabling the monitoring of the optical evolution of the system[40,41]. Figure 4h shows the dark-field optical image of a single crack propagating from a drying droplet. Figure 4j displays the plasmon resonance curves as a function of wavelength and of the distance from the colloidal solution to the dried film (noted from A to J). Figure 4i summarizes the spatial evolution of the wavelength corresponding to the plasmonic peak. The solution exhibits a plasmonic peak at around 835 nm (zone A). During drying, cracks nucleate when a colloidal gel film filled with water is formed. The formation of the colloidal "wet" gel leads to a shift of the plasmonic peak to 905 nm (zone J), attributed to the higher refractive index of PS colloids ($n_{PS} = 1.59$) surrounding the AuBP compared to water ($n_{H2O} = 1.33$). No characteristic plasmonic signals could be recorded in the zone corresponding to the dried

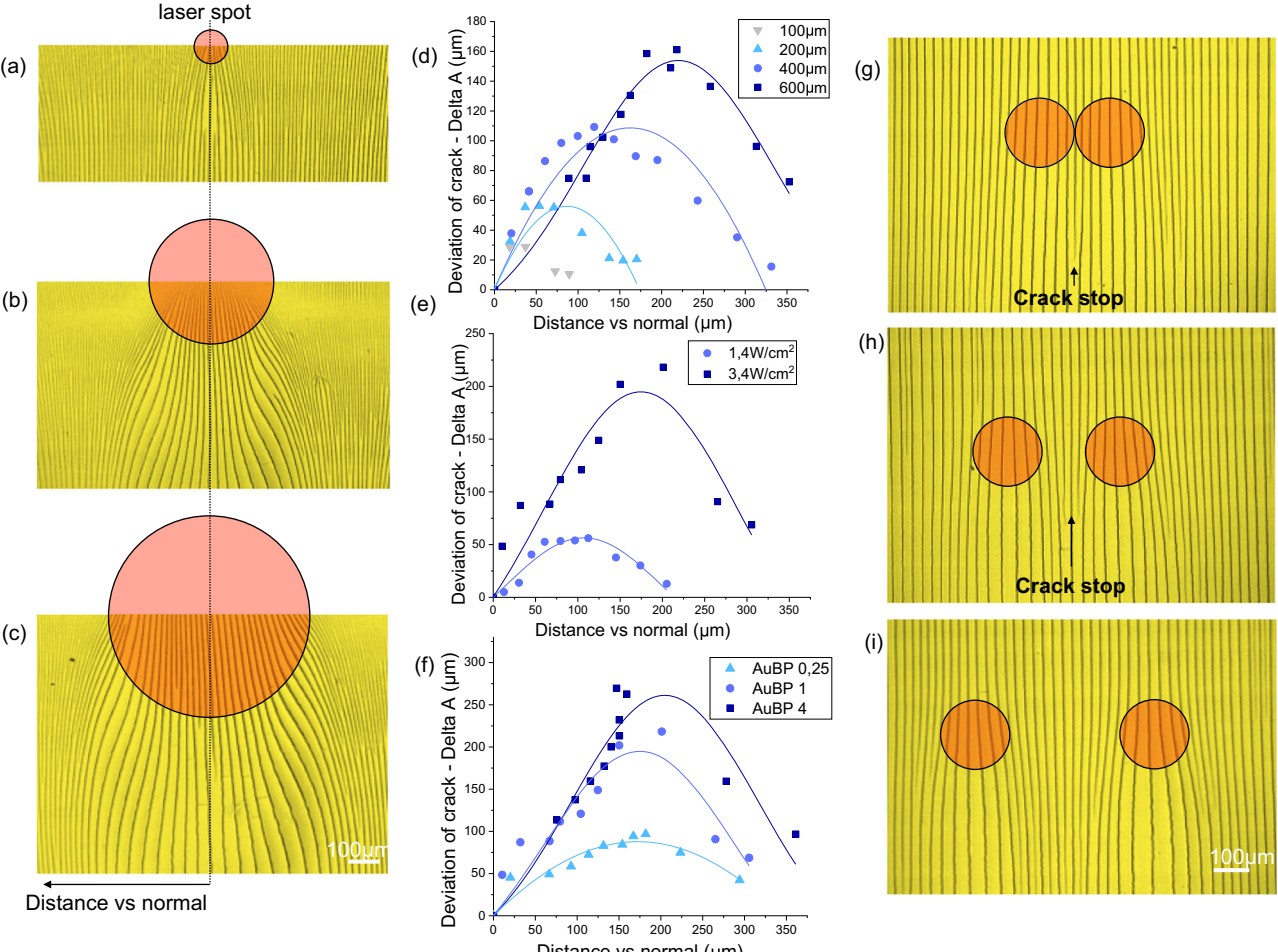

**Fig. 5 | Analysis of the deviation of cracks. a, b, c** Optical microscopy images of deviated cracks due to a circular laser spot of 100 μm, 400 μm and 600 μm respectively. **d** Analysis of the cracks deviation – delta A – depending on the size of the hole of the mask. **e** Analysis of the cracks deviation – delta A – depending on the laser's power. **f** Analysis of the cracks deviation – delta A – depending on the concentration in AuBP. Specifically, the concentration of plasmonic particles was multiplied by four (AuBP 4) and reduced by four (AuBP 0.25), while keeping the other conditions constant (laser power of 3.4 W/cm² and spot size of 400 μm). Lines correspond to the Gaussian fit. **g, h, i** Optical microscopy images of deviated cracks due to dimer spots spaced of 0 μm, 100 μm and 300 μm respectively. Source data of Figure (**d, e, f**) are provided as a Source Data file.

colloidal film (brighter upper part of Fig. 4h) due to the strong scattering of the PS colloids once the water is completely evaporated. By irradiating the surface with an 852 nm laser (as shown in Fig. 4a), the optimal photo-heating effect will be achieved, in principle, where the maximum of the plasmonic absorption peak corresponds to 852 nm that is predominantly located in the solution, in the vicinity of the gelification front.

### Analysis of crack deviation

To control the process and program crack patterns, a detailed analysis of the extent of deviation as a function of the spot size, laser power and AuBP concentration was conducted. Figure 4c illustrates the two primary features that we analyzed through image analysis (Supplementary Fig. 16): the angle of deviation (α) and the distance of deviation in relation to the normal (delta A). The effect of the hole size in the mask was investigated while keeping the laser power constant (2.1 W/cm²). Micrographs of the crack patterns obtained with different hole diameters of 100 μm, 400 μm, and 600 μm are displayed in Fig. 5a, b, c, respectively. This trend is further illustrated in the plots in Fig. 5d, representing the crack deviation and angle as a function of distance versus normal for different spot sizes. The trends can be fitted with a Gaussian function. In all cases, a minimal deviation is observed at the center of the spot. Instead, maximum deviation is reached at a certain

distance from the center of the irradiation spot, which increases with the spot size. This can be attributed to larger laser spots resulting in a greater amount of evaporated water, leading to a larger deformation of the drying line and forming a concave profile. As expected, higher power or higher concentration of AuBP results in a greater local temperature, leading to larger crack deviations (Fig. 5e, f). The same trends were observed for the angle of deviation, as shown in Supplementary Fig. 17. Similarly, we conducted an experiment to demonstrate the wavelength-dependency in crack deviation (Supplementary Fig. 18) by irradiating the Au BPs with lasers at different wavelengths (532 and 852 nm) while maintaining the same power of 3.4 W/cm². As shown in Supplementary Fig. 18a and b, light at 852 nm is strongly absorbed (longitudinal plasmonic peak) leading to a maximum heating (T > 100 °C). In contrast, irradiation at 532 nm was less efficient, resulting in a temperature of approximately 40 °C. This reduced efficiency can be attributed to the weaker absorption of the transverse plasmonic peak. For reference, we compared these temperature values with the temperature of a system without Au bipyramids, which did not exhibit significant heating and deviation. When applied to the light-induced dip-coating experiment for deviating the cracks, 532 nm irradiation results in a lower deviation compared to the 852 nm, as shown in Supplementary Fig. 18c. This experiment confirms that exciting plasmonic particles off-resonance reduces photon

absorption, heat conversion, and, consequently, crack deviation. As a consequence, without AuBPs, no deviation of cracks is observed (Supplementary Fig. 19). This analysis focuses on the deviation of cracks from single spots. Subsequently, we investigated the effect on crack deviation in the case of light "dimers," consisting of two adjacent circular spots with different gaps between the spots. Figure 5g, h, i display micrographs of the cracked colloidal film obtained by exposing light dimers (spot size of 200 μm) with gaps of 0 μm, 100 μm, and 300 μm, respectively. Two distinct behaviors can be observed. For a large gap (300 μm, Fig. 5i), each light spot independently induces crack deviation. Conversely, as the gap decreases (Fig. 5g, h), "self-arresting" of some cracks is observed between the two spots due to the convergence of cracks that are too close to each other. This behavior is contrary to the "self-replication" of cracks mentioned earlier and observed in Fig. 1d, where new cracks appear when two adjacent cracks diverge. An image analysis was conducted to evaluate when a crack forms or arrests as a function of the divergence or convergence ratios of two adjacent cracks, as depicted in Supplementary Fig. 20. On average, self-replication is observed for a divergence of 0.46, while self-arresting of a crack occurs for a convergence of 0.74.

## Discussion

Starting from the aforementioned analysis, it is possible, in principle, to program the formation of crack patterns and anticipate the trajectory of cracks in terms of angle of deviation, self-replication, and self-arresting, as explained in Supplementary Fig. 21. From a given light pattern, we introduce a simple graphical approach to design the "expected" final crack pattern. Briefly, the process involves several steps. The first step involves the design of deviated cracks as a function of their distance from the light spot. The extent of deviation for each crack is determined by the experimental values of Delta A and α, as previously identified. This results in the formation of an array of deviated cracks (Supplementary Fig. 21d). As discussed earlier, when two adjacent cracks come too close, "self-arresting" occurs. To account for this, the second step involves arresting the cracks when the convergence ratio reaches 0.74, as depicted in Supplementary Fig. 21f. The outcome of this simple process is referred to as the "expected" crack pattern and can be compared with the "experimental" crack pattern obtained after the actual experiment. Two examples of "expected" and "experimental" crack patterns are presented in Fig. 6b and Supplementary Fig. 22a, demonstrating a notable agreement between the two patterns. To further expand the capabilities, larger undulating patterns can now be programmed by irradiating the surface with arrays of light beams, as illustrated in Fig. 6a, c, d, and Supplementary Fig. 22b, c, showcasing various examples of curved patterns with distinct morphologies. These examples involve films composed of colloidal PS and AuBP, as depicted in the SEM micrograph in Fig. 6e. To further prove the versatility of this light-directed patterning method, we extended the approach to other materials. As a first example, the patterned colloidal film can be utilized as a mask to fabricate arrays of gold lines on glass via lift-off, as shown in Fig. 6f, g, and Supplementary Fig. 23, opening up intriguing possibilities for transparent electrodes as reported previously[12,17]. A second example is shown in Fig. 6h, i demonstrate the fabrication of light-directed patterned films made of porous $TiO_2:SiO_2$, achieved by incorporating titania and silica sol-gel precursor into the colloidal plasmonic solution and subsequently annealing the film at 500 °C after deposition[25]. At last, in principle, the crack self-ordering process can be generalized to other colloidal solutions with different compositions, sizes and solvents. The overall quality of the periodic patterns and the presence of defects vary significantly from one system to another and the characteristics of the colloidal solution (concentration, stabilizer, etc.) need to be optimized on a case-by-case basis. As an example, Supplementary Fig. 24 displays oriented crack patterns in colloidal films made of YAG:Ce, $SiO_2$ and PMMA colloids of different sizes obtained by drying an aqueous

colloidal droplet. Using larger particles of a size comparable to the wavelength of the laser would open up very interesting possibilities for generating additional optical features. For instance, larger monodispersed colloids can be self-assembled reversibly into 2D or 3D photonic crystals, giving rise to structural colors[42]. As an intriguing perspective for our work, the colloidal self-assembly process and the crack formation could be monitored by tracking the optical evolution of the photonic colloidal structures. In addition, using even larger-sized colloids would enable even more sophisticated interactions between light and the colloidal assembly, such as multiple scattering, optical amplification of light such as reconfigurable random lasers[43].

In conclusion, we have demonstrated that the propagation of cracks in solution-processed colloidal films can be manipulated by light. By utilizing plasmonic photothermal absorbers, we were able to create 'virtual defects' in the form of programmable heating zones. Drawing an analogy to negative phototropism in plants, we have shown that propagating cracks can either evade light, self-replicate, or arrest, allowing for the generation of curved patterns through light-directed self-ordering of cracks. To achieve this, we engineered the process by applying light irradiation during dip-coating, enabling simultaneous deposition and light-directed crack patterning of different materials. This approach represents a rare example of a robust self-assembly process that can be programmed in both space and time, resulting in long-range order. While mainly driven by scientific curiosity, this light-driven control in crack propagation may inspire the development of crack-related failure mitigation strategies in various fields, including structural engineering.

## Methods

### Fabrication of the PS nanoparticles

The synthesis was performed in a 2.5 L jacket glass reactor with an air-tightly connected mechanical stirrer and 4 necks. Briefly, the dissolution of 40 g Pluronic F127 (Sigma Aldrich) in 1600 mL of MiliQ water was followed by the addition of 240 g of styrene monomer (Sigma Aldrich) and 3.46 g of $NaHCO_3$ (Alfa Aesar). The monomer was used as received, without any additional purification. The solution was stirred and flushed with $N_2$ for 1 h in order to remove air. In parallel, the solution of the initiator made of potassium persulfate (3.60 g from Sigma Aldrich) dissolved in an additional 80 mL of water was flushed with $N_2$ for 30 min. The initiator solution was subsequently added to the monomer solution and stirred and flushed with $N_2$ for another 20 min. The mechanical stirrer was set throughout the process at 270 rpm. The mechanical stirrer was air-tight connected and a slight $N_2$ flush was kept over the solution level in order to provide slight overpressure, released by a needle incorporated into the rubber cap of one neck. Then, the temperature of the circulated liquid in the jacket was raised to 90 °C (achieving 70 °C after 35 min and 90 °C after 55 min). From the liquid reached 90 °C the reaction was continued for 5 h. After that time the temperature was set to 40 °C and all the valves were opened in order to quench the radicals with oxygen. The obtained milky solution was further centrifuged at different speeds and separated from the precipitate. The utilized speeds were consecutive: 1) 1500 rcf × g for 10 min; 2) 4500 rcf × g for 20 min; 3) 6000 rcf × g for 15 min; 4) 9500 rcf × g for 40 min. The final solution was dialyzed with MiliQ water exchanged 6 times each 12 h. After that stable colloidal solution of PS latex particles was obtained. The solution could be further concentrated to the desired concentration by a rotatory evaporator.

### Fabrication of the titania - silica solution

First a titania solution is prepared. 0.75 g of Acetylacetone (acac) (Sigma Aldrich) was added dropwise into titanium n-butoxide (1 g). The solution was stirred for 20 min, after which ethanol (2.16 g) was added (in order to facilitate mixing with aqueous-based solutions, which is otherwise immiscible). Then, the solution was stirred for 1 h and filtered through a nylon syringe filter (0.45 μm). The obtained

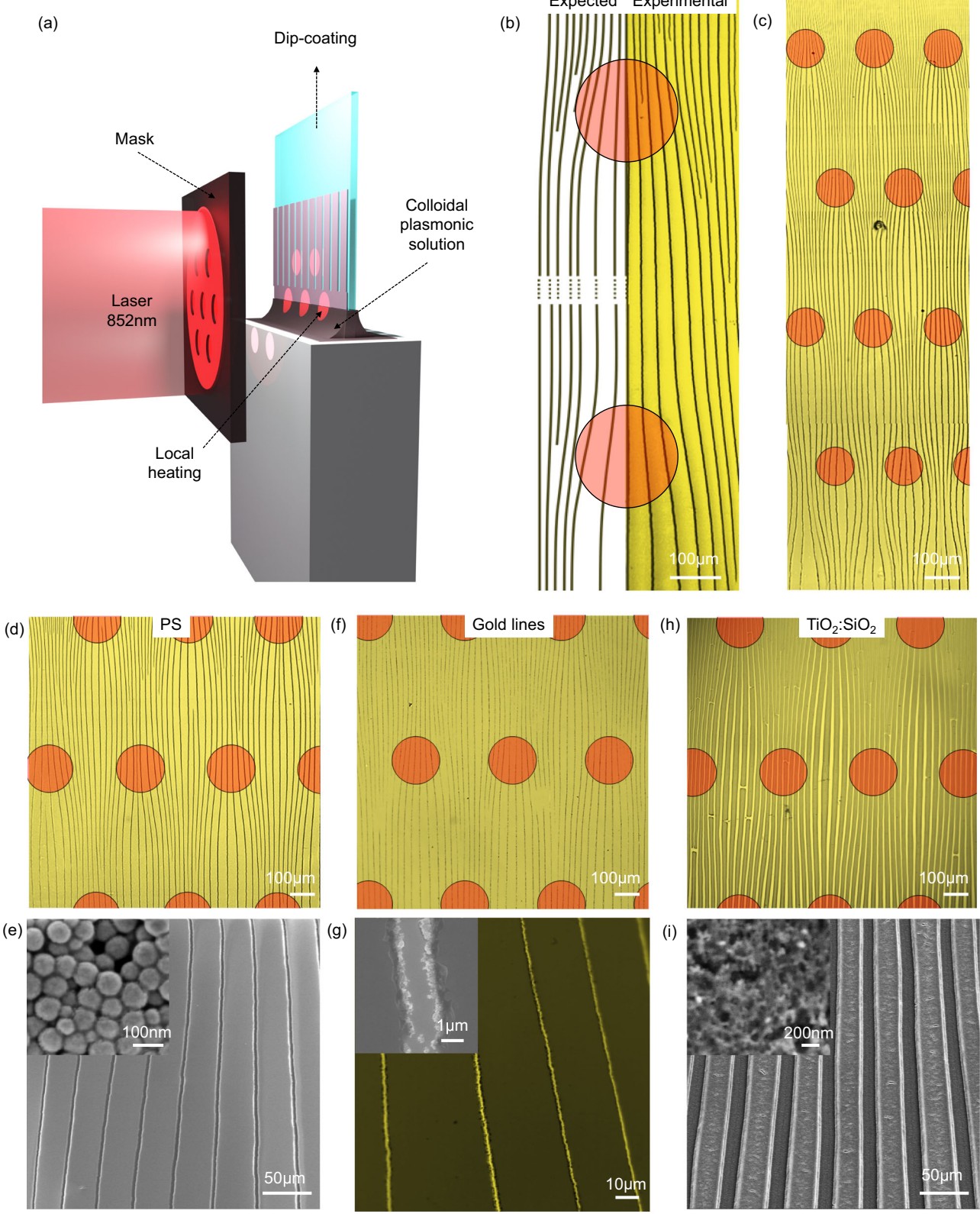

**Fig. 6 | Examples of curved patterns with distinct morphologies. a** Scheme of the experimental set-up. **b** Expected and experimental pattern obtained for two circular spots of 200 μm diameter. Red circles represent the position of the spots of laser. **c**, **d** Optical micrographs of patterns of deviated cracks. **e** SEM micrograph of the pattern and the polystyrene nanoparticles in the inset. **f** Optical micrographs of a gold lines pattern, with **g** the corresponding SEM image. **h** Optical micrograph of a porous TiO$_2$:SiO$_2$ curved pattern with **i** the corresponding SEM image. The inset shows the porous structure.

yellow/orange solution is denoted in the manuscript as the titania precursor. 5 g of PS latex solution (16 wt% of solid) was added dropwise (using a syringe pump; 1 drop per 5–10 s) under magnetic stirring into the titania precursor (0.5 g) and stirred overnight. The solution was centrifuged at 5500 rcf × g for 7 min, to remove the small amount of yellow precipitate. The final yellow solution was stable for months. Then, 2,1 g of titania solution was mixed with TEOS (0.17 g).

### Fabrication of gold bipyramids Au@840 nm

The nanobipyramids and their seeds were prepared using the concentrated procedure[39]. Tetrachloroauric acid trihydrate (99.99%), tetramethylguanidine (99%) and Catechol (98%), were procured from Alfa Aesar, while Citric acid monohydrate (99.5%), CTAC (25% in water), CTAB (99%, ref. H6269), and PEG2000-SH were obtained from Sigma-Aldrich. All vials, glassware, and magnetic stirrers underwent thorough cleaning with regal water and extensive rinsing in MilliQ water before usage. Magnetic stirrers were meticulously rinsed by sonicating them for a few minutes in MilliQ water twice, followed by storage in MilliQ water. Solutions of catechol, particularly tetramethylguanidine (TMG), were prepared immediately before use. $NaBH_4/NaOH$ solutions, stored in the fridge at 5 M, were diluted to 50 mM in ice-cold water just before application.

**Preparation of bipyramid seeds and pre-growth.** 6.6 g of commercial 25% CTAC (0.78 M) were added to 73 g of MilliQ water, in a 250 mL round-bottom flask, equipped with a 40 × 20 mm oval stirring bar, followed by the addition of 800 µl of 25 mM HAuCl4 and 1 mL of 0.1 M citric acid solution. The resulting mixture was stirred until complete transparency was achieved and then cooled to 19–21 °C. Then, 1 mL of 50 mM $NaBH_4/NaOH$ solution was added in 2 s, under vigorous stirring (900 rpm), causing the color to shift from pale yellow to a brownish orange. 600 µL of 0.1 M citric acid solution were added, after one minute of continuous stirring, and the stirring rate was reduced to 200–300 rpm. The flask was heated to 85 °C in a preheated oil bath for 120 min. After the removal of the oil bath the seed solution was allowed to cool gradually.

As the seed solution reached a temperature of 45–50 °C, the stirring speed was elevated to 600–700 rpm, and a preheated (40 °C) growth solution, composed of 4.4 mL of 25 mM $HAuCl_4$, 800 µL of 40 mM $AgNO_3$, 38 mL of 140 mM CTAB, 1.1 mL of 0.1 M NaOH, and 2.4 mL of 0.4 M ethanolic HQL (added at the last minute), was added into the seed flask. The mixture was maintained between 45 and 50 °C for 40 min using a water bath. The pre-grown seeds were then centrifuged for 45 min at 13000 rcf × g, the supernatant (pale orange-red) was discarded, and the particles were dispersed in 6.5 mL of 1 mM CTAB to yield a highly concentrated seed suspension.

**Growth of the bipyramids and functionalization with PEG2000-SH.** 130 µL of $AgNO_3$ solution (40 mM), 7.5 mL of 137 mM CTAB and 1000 µL of concentrated overgrown seeds were added in a 50 mL round bottom flask and stirred for 5 min in a water bath at 60 °C. 5 mL of 100 mM $HAuCl_4$ solution was added quickly under vigorous stirring to a mixture of 15 mL of 137 mM CTAB, in a 50 mL beaker followed by 900 µL of an aqueous solution of tetramethylguanidine (0.8 M). After 1–2 min of stirring, the solution became homogeneous, and 320 µL of $AgNO3$ solution (40 mM) was added followed by 3.3 mL of 0.8 M catechol solution in 25:75 $EtOH/H_2O$. The mixture was stirred for 2–3 min, during which the color transitioned from orange to deep brown.

Then, this growth solution was gradually added to the hot solution in the round bottom flask in 4 min under vigorous stirring using a pipette. The heating was maintained for an additional 15 min. At the end, 50 µL of $AgNO_3$ solution (40 mM) were added and the heating was maintained for an extra 5 min. The resulting particles were cooled to 45 °C and centrifuged with the addition of 3 mL of EtOH (to reduce the viscosity) at 7000 rcf × g, and then redispersed in 35 mL of 5 mM CTAB

using a sonic probe. The particles were stored for one night at RT, and then centrifuged again at 4000 rcf × g. The particles were then redispersed in 35 mL of a 1 mM (2 g/l) PEG2000-SH solution and incubated overnight. Another round of centrifugation at 4000 rcf × g ensued, and the particles were redispersed in 20 mL of MilliQ water with the addition of 5 mg of PEG2000-SH, resulting in a stable concentrated suspension of gold bipyramids (AuBP@840 nm PEG 2000 at 4.5 g/L in [$Au^0$]).

### Fabrication of the dotted plasmonic substrates

A glass substrate was washed with acetone and placed under plasma (plasma cleaner PDC-32G-2) for a few seconds. A solution of AZ 1512 HS photoresist was spin-coated (SCS-6800 Spin Coater Series) on the glass slide (4000 rpm – 20 s). The resist film was illuminated (25 s) patterned by a mask-less SmartPrint device (Smart Forces Technologies). The slide was then washed in the developer solution (AZ 400 K Developer) (10 s) and rinsed abundantly in water.

Gold nanoparticles by dewetting: after photopatterning, a 10 nm gold layer was then applied by sputtering over the glass slide. The sample was then sonicated in acetone for one minute to remove the top gold layer and obtained the final pattern. Finally, the glass slide was heated at 450 °C for 20 min to form pseudo spherical particle by dewetting of the remaining gold layer.

Gold bipyramids by drop-casting: after photopatterning, a droplet of AuBP solution was applied on the patterned substrate for one minute and then rinsed with distilled water. Finally, the resist was removed with acetone to reveal the pattern.

### Light-driven dip-coating

The deposition was carried out by dip-coating (ACEdip®, Solgelway) under atmospheric humidity and temperature. In a typical process, the substrate was immersed in the colloidal solution at the constant speed of 2 mm s$^{-1}$ and withdrawn speeds 0.004 mm s$^{-1}$ with a PS concentration of 17 wt%. The laser at 532 nm was a LRS-0532-PF-01000-01 laser provided from Laserglow Technologies. The laser at 852 nm was a MDL-III-852nm-1W laser provided by Changchun New Industries Optoelectronics Tech. Co.

### Characterization

Optical and hyperspectral microscopy. Optical pictures and hyperspectral images were acquired by optical microscope (Cytoviva). Hyperspectral image of the evolution of the plasmonic scattered peak was obtained in dark field mode by push-broom method with an acquisition time of 2 s, using a ×10 objective (reflection mode). The evaporation of the droplet was controlled by a heating/cooling stage (Linkam Scientific) fixing the temperature at 30 °C. Scanning electron microscopy (SEM) imaging was performed with SU-70 Hitachi FESEM, equipped with a Schottky electron emission gun. The SEM cross-section micrographs were obtained by cleaving the sample in the vicinity of the deviated cracks and observing it perpendicularly to the substrate plane using SEM. The analysis of the elemental composition (EDX) was performed on a ZEISS Gemini SEM 360 equipped with an Oxford Instruments Ultim Max 170 mm$^2$ detector. SEM images and EDX mapping were obtained by an Inlens SE detector (in column). Oxford Instrument Aztec software was used for the acquisition of EDX maps. Infrared measurement was recorded in the range of 4000 and 550 cm$^{-1}$ on a Perkin Elmer Spectrum 400 spectrometer.

## Data availability

Optical spectra and data analysis from the main text or the Supplementary Information are provided with this paper. Source data are provided with this paper.

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

## Acknowledgements

We thank D. Montero and the Institut des Matériaux de Paris Centre (IMPC FR2482) for servicing FEGSEM & EDX instrumentation and Sorbonne Université, CNRS and C'Nano projects of the Région Ile-de-France for funding. This work was supported by the European Research Council (ERC) under European Union's Horizon 2020 Programme (Grant Agreement no. 803220, TEMPORE).

## Author contributions

M.F., M.O. and F.T. performed the experiments on drying droplets. F.T. performed the light-assisted dip-coating experiments. D.C. and S.P. synthetized the AuBPs. M.O. synthetized PS particles. F.T. performed the SEM and hyperspectral characterizations. F.T. and M.F. wrote the manuscript. All the authors read and gave constructive inputs to the manuscript. M.F. conceived the idea and supervised the project.

## Competing interests

The authors declare no competing interests.
