## [Peer Review File · Nature Communications]

Programming crack patterns with light in colloidal plasmonic filmsREVIEWER COMMENTS

Reviewer #1 (Remarks to the Author):

The authors report on a novel approach for patterning colloidal layers by controlling the propagation and nucleation of cracks during drying of the films. The key idea is to locally vary temperature in the drying films using photo thermal heating effects that can be localized via plasmonic nanoparticles. The local heating changes the evaporation rate and thus distorts the drying front and consequently influences the crack propagation. The authors demonstrate the effect both for droplet drying and in a more defined setting for crack formation in a dip coating scenario. They very convincingly illustrate that the effect can be triggered by homogenous illumination of patterned substrates or by patterned illumination of homogenous films in which either the substrate can be heated plasmonically or the plasmonic particles are incorporated in the colloidal suspension. The analysis of the underlying phenomena is compelling. As well as the variety of patterning variations illustrates that this is a versatile method. It is a beautiful and elegant work of high quality.

The only point that remains not fully clear to me is how general this mechanism is for other colloidal systems than the particular one under investigation here. In my experience crack formation in drying colloidal layers typically is not as homogenous and following a strict periodicity as in this example. How is random crack nucleation due to defects suppressed? Is this behaviour specific for the case of pluronic modification, can it be generalized to other particle types beyond PS particles? The authors focus very much on manipulation of the crack formation and take the well defined periodic crack pattern for granted. The latter however is the prerequisite for the control. So it would be important to learn more about how easy it is to achieve such well defined crack formation for other colloidal layers.

Reviewer #2 (Remarks to the Author):

The manuscript presents a novel and intriguing approach to control the propagation of cracks in colloidal films using light. The experiments are well-designed, and the results are supported by the data presented. The role of gold particles and their plasmonic properties in influencing crack propagation is particularly interesting. However, a more detailed explanation of the relationship between the plasmon wavelength and irradiation, as depicted in Figure 3, would enhance the clarity of the findings.

The role of the PS particles is also somewhat secondary in this study and it would be interesting to know how the size of these particles influences the experiments. It is especially interesting to know what would happen with even larger particles, of a size comparable to the wavelength of the laser used, and much more monodisperse (from the images it seems that the ones used in the experiments are not) since it would give the possibility to study self-assembly phenomena and their possible manipulation. In addition, even larger particles (more than one micron) could resonate with the lasers used (through Mie modes) and introduce even more possibilities.

I would like the authors to discuss this in the final version of the paper and consider the possibilities beyond crack control.

Reviewer #3 (Remarks to the Author):

The authors present an interesting study in which they show how the interplay between visible light laser illumination, plasmonic nanoparticles and a colloidal suspension can be harnessed to steer the growth direction of cracks in a forming colloidal thin film on a surface. As the underlying mechanism they propose the local heating via light absorption and dissipation in either Au plasmonic particles on the colloidal film substrate or dispersed directly in the colloidal suspension.

In general, this study is well executed with a suite of experiments that test and verify the proposed

mechanism in a convincing and careful way. From this perspective, as well as the perspective of novelty this work has the potential to be of interest for the readership of Nature Communications. However, before any decision can be made, the authors should address the following issues:

- The English should be improved quite generally throughout the entire text, as well as numerous typos should be eliminated by careful proof-reading.
- In both introduction and conclusions section, I feel that the authors are somewhat overselling their results by stating relevance for certain types of applications. Quite honestly, it is not obvious to me how the observed effect and approach described in this work really would be relevant/applicable the way it is proposed (e.g. it is not at all obvious to me how this method "opens exciting possibilities for bottom-up device fabrication". What kind of devices? In which area? In which way are cracks important in devices, etc.). Hence, I suggest that the authors either further elaborate in detail how they envision such applications to happen or simply reduce their claims and focus on highlighting the more fundamental – very interesting (!) – aspects of their work. I would argue that the latter should be (more than) enough. Also fundamental curious discoveries deserve to be in the spotlight!
- In the context of Figure 1, the general arrangement of the sample could be better explained graphically, e.g., how the plasmonic substrate is localized with respect to the drying front of the colloidal film and how/where exactly the drying droplet is illuminated and imaged. Right now, I am guessing (?) that it is at the edge of the droplet, but it really is not clear.
- Related to the point above, in Figure 1c, where are these images taken? At the edge of the droplet?
- The authors are describing the use of "Au spheres". I would argue that such terminology is incorrect because metallic colloidal nanocrystals rarely are spherical but actually faceted crystals. Hence the word "sphere" is misleading.
- When the Au "spheres" and bipyramids are introduced, they are not characterized in terms of their plasmonic properties and it is not motivated why two different types are used. This becomes clear only much later in the text when the different LSPR wavelengths are mentioned and there is referral to figures in SI. It would help the reader to understand this work if, at first mention, the different plasmonic properties of the two types of Au NPs used would be characterized (e.g. by an extinction spectrum of the colloidal solution) and shown, together with a short explanation of what can be achieved by having these two types of particles in terms of light manipulation/interaction.
- In the context of Figure 2g, it is demonstrated that changing the arrangement and size of the plasmonic dots can be used to tune the crack pattern. Then it is also mentioned that the same effect can be obtained using colloidal NPs of two different types, for which laser wavelengths matching their respective LSPRs are used. Here, it would be relevant to showcase also control experiments where the respective samples are illuminated also off-resonance, i.e. the "spheres" with 852 nm and the bipyramids with 532 nm. An even more elegant and convincing experiment that I highly recommend is actually execute a scan across multiple wavelength below, at, and above the LSPR of each type of particle, to really reveal the importance of LSPR excitation.
- In Figures 2 (h) and (i) the small bipyramid symbols are to illustrate different orientations according to the caption but they are oriented in the same way in the two panels?
- The experiments depicted in Figure 3 show how the crack formation in the drying colloidal film is impacted by laser illumination at the LSPR wavelength of Au bipyramids dispersed in the colloidal suspension. While the effect is clear, what I am lacking is an identical control experiment without Au nanoparticles in solution.
- Figure 3 (g) is explained as cross-sectional "micrograph". However, it is not explained how it was obtained. Furthermore, in the caption "hyperspectral analysis" is mentioned. What is meant by this? In which way is this micrograph "hyperspectral"?
- Figure 4 d, e, f in particular (but in many other figures as well) font sizes used are very small. Please improve overall readability of figures in this respect
- The explanations given in Figure S19 are very important and I therefore suggest to actually include this in the main text.
- On line 286 (as well as in in SI Figure 20) the authors mention "predicted" crack patterns. How are they predicted? Is any kind of model used? This is unclear and nowhere explained?!
- On line 295 the authors mention "transparent" electrodes. It is unclear to me in which way this

method could be used to make transparent electrodes when they are made from Au lines (which indeed interact with visible light?). Furthermore, what would be the advantage over making such electrodes simply with e.g. photolithography?

We would like to thank the reviewers for the time spent to evaluate our work and for their feedback and constructive comments. Please find hereafter our point-by-point response; in addition the modifications in the manuscript are highlighted in yellow.

REVIEWER COMMENTS

Reviewer #1 (Remarks to the Author):

The authors report on a novel approach for patterning colloidal layers by controlling the propagation and nucleation of cracks during drying of the films. The key idea is to locally vary temperature in the drying films using photo thermal heating effects that can be localized via plasmonic nanoparticles. The local heating changes the evaporation rate and thus distorts the drying front and consequently influences the crack propagation. The authors demonstrate the effect both for droplet drying and in a more defined setting for crack formation in a dip coating scenario. They very convincingly illustrate that the effect can be triggered by homogenous illumination of patterned substrates or by patterned illumination of homogenous films in which either the substrate can be heated plasmonically or the plasmonic particles are incorporated in the colloidal suspension. The analysis of the underlying phenomena is compelling. As well as the variety of patterning variations illustrates that this is a versatile method. It is a beautiful and elegant work of high quality.

Response: we thank the referee for the positive feedback. Below we address the remaining questions.

The only point that remains not fully clear to me is how general this mechanism is for other colloidal systems than the particular one under investigation here. In my experience crack formation in drying colloidal layers typically is not as homogenous and following a strict periodicity as in this example. How is random crack nucleation due to defects suppressed? Is this behaviour specific for the case of pluronics modification, can it be generalized to other particle types beyond PS particles? The authors focus very much on manipulation of the crack formation and take the well defined periodic crack pattern for granted. The latter however is the prerequisite for the control. So it would be important to learn more about how easy it is to achieve such well defined crack formation for other colloidal layers.

Response: we agree with the point raised by the referee regarding the generality of the crack self-ordering process and its applicability to other colloidal systems. In principle, the **crack self-ordering process can be generalized to other colloidal solutions** with different compositions, sizes and solvents. To support this statement, we provide several micrographs showing oriented crack patterns in colloidal films made of YAG:Ce, SiO₂ and PMMA colloids, obtained by drying an aqueous colloidal droplet.

In these cases, we observe radially aligned cracks similar to those shown for PS in Figure 1 and Figure S3. However, we also need to specify that the overall quality of the periodic patterns and the presence of defects vary significantly from one system to another. At this stage, to the best of our knowledge, there is no general rule, and the characteristics of the colloidal solution (concentration, stabilizer, etc.) need to be optimized on a case-by-case basis. Based on that, PS particles give the best results in terms of quality and ordering that this is the reason why, we focused on this system.

It is indeed important to discuss this point.

ACTION TAKEN

To reflect the above considerations, we have included in this revised version:

- a discussion at page 14 on how the crack self-ordering process can be generalized to other colloidal systems
- the micrographs with YAG:Ce, SiO₂ and PMMA in SI as Figure S24

Reviewer #2 (Remarks to the Author):

The manuscript presents a novel and intriguing approach to control the propagation of cracks in colloidal films using light. The experiments are well-designed, and the results are supported by the data presented.

Response: we thank the referee for the positive feedback. Below we address the remaining questions.

The role of gold particles and their plasmonic properties in influencing crack propagation is particularly interesting. However, a more detailed explanation of the relationship between the plasmon wavelength and irradiation, as depicted in Figure 3, would enhance the clarity of the findings.

Response: we thank the reviewer for the suggestion. To clarify the relationship between plasmon wavelength and irradiation we have taken two actions.

ACTIONS TAKEN:

1. We added Figure 1(e) to illustrate the ideal relationship between plasmon wavelength and laser wavelength. In addition the following sentences have been added in page 6: "*We attributed the origin of the light directed self-ordering to evaporation driven photo-thermal effect. To achieve high efficient photo-heating, the laser wavelength ideally needs to match the absorbing specific wavelengths on the plasmonic nanoparticles as illustrated in Figure 1(e).*"³³ *When the nanoparticles are exposed to laser irradiation at their resonance frequency, a maximum of energy from the absorbed photons is converted into heat. Irradiation at different resonances results in lower photothermal heating."*

2. In line with what was also requested by Reviewer 3, we conducted an experiment to demonstrate the wavelength-dependent crack deviation, especially for the Au BPs system. When exciting plasmonic

particles off-resonance, we basically expect a reduced or negligible absorption of photons and conversion into heat.

We characterized the temperature increase using an IR camera and analyzed crack deviation through image analysis. The Au bipyramids, irradiated with the different lasers, exhibit a wavelength depend crack deviation as shown in the following Figure (S18).

Figure (a) illustrates the absorbance spectrum of colloidal Au bipyramids, which exhibit two absorption peaks. The spectrum displays a prominent longitudinal absorption peak at around 850 nm and a weaker transverse absorption peak at approximately 515 nm.

To assess the photothermal heating effect, we irradiated the colloidal system with lasers at different wavelengths (532 and 852 nm) while maintaining the same power of 3.39 W/cm². As shown in Figure (b), irradiation at 852 nm led to the maximum heating, reaching temperatures exceeding 100°C. In contrast, irradiation at 532 nm was less efficient, resulting in a temperature of approximately 40°C. This reduced efficiency can be attributed to the weaker transverse absorption peak and the mismatch with the resonance wavelength of the transverse absorption peak.

For reference, we compared these temperature values with the temperature of a system without Au bipyramids, which did not exhibit significant heating. When applied to the light-induced dip-coating experiment for deviating the cracks, 532nm irradiation results in a lower deviation compared to the 852nm, as shown in Figure (c). In summary, we confirm that exciting plasmonic particles off-resonance reduces photon absorption, heat conversion, and, consequently, crack deviation.

The role of the PS particles is also somewhat secondary in this study and it would be interesting to know how the size of these particles influences the experiments. It is especially interesting to know what would happen with even larger particles, of a size comparable to the wavelength of the laser used, and much more monodisperse (from the images it seems that the ones used in the experiments are not) since it would give the possibility to study self-assembly phenomena and their possible manipulation. In addition, even larger particles (more than one micron) could resonate with the lasers used (through MIE modes) and introduce even more possibilities. I would like the authors to discuss this in the final version of the paper and consider the possibilities beyond crack control.

Response: We thank the reviewer for this suggestion. We agree that using larger particles of a size comparable to the wavelength of the laser would open up very interesting possibilities for generating additional optical features. For instance, as mentioned by the reviewer, larger monodispersed colloids can be self-assembled reversibly into 2D or 3D photonic crystals, giving rise to structural colors (J. Am. Chem. Soc. 2021, 143, 30, 11535–11543). As an intriguing perspective for our work, the colloidal self-assembly process and the crack formation could be monitored by tracking the optical evolution of the photonic structure.

In addition, using even larger-sized colloids would enable even more sophisticated interactions between light and the colloidal assembly, such as multiple scattering, optical amplification of light resulting reconfigurable random lasing (Nat. Phys. 2022, 18, 939–944). We anticipate that these are indeed some of our short-term perspectives for a follow-up article.

ACTION TAKEN

These perspectives are now addressed in the revised manuscript on page 16.

Reviewer #3 (Remarks to the Author):

The authors present an interesting study in which they show how the interplay between visible light laser illumination, plasmonic nanoparticles and a colloidal suspension can be harnessed to steer the growth direction of cracks in a forming colloidal thin film on a surface. As the underlying mechanism they propose the local heating via light absorption and dissipation in either Au plasmonic particles on the colloidal film substrate or dispersed directly in the colloidal suspension.

In general, this study is well executed with a suite of experiments that test and verify the proposed mechanism in a convincing and careful way. From this perspective, as well as the perspective of novelty this work has the potential to be of interest for the readership of Nature Communications.

Response: we are delighted that the referee believes our work is of potential interest for publication in Nature Communications and are thankful for the constructive comments. Below we address the remaining questions.

However, before any decision can be made, the authors should address the following issues:

- The English should be improved quite generally throughout the entire text, as well as numerous typos should be eliminated by careful proof-reading.

Response: thank you for the suggestion, we have revised our manuscript to remove typos and rephrase some sentences.

- In both introduction and conclusions section, I feel that the authors are somewhat overselling their results by stating relevance for certain types of applications. Quite honestly, it is not obvious to me how the observed effect and approach described in this work really would be relevant/applicable the way it is proposed (e.g. it is not at all obvious to me how this method "opens exciting possibilities for bottom-up device fabrication". What kind of devices? In which area? In which way are cracks important in devices, etc.). Hence, I suggest that the authors either further elaborate in detail how they envision such applications to happen or simply reduce their claims and focus on highlighting the more fundamental – very interesting (!) – aspects of their work. I would argue that the latter should be (more than) enough. Also fundamental curious discoveries deserve to be in the spotlight!

Response: thank you very much for the comment; we couldn't agree more with the reviewer's view. Indeed, even though this method may have implications for the fabrication of patterned surfaces, our work is fundamentally driven by scientific curiosity.

ACTION TAKEN

In line with the reviewer's suggestion, we have removed comments about applications from the abstract, introduction, and conclusions.

- In the context of Figure 1, the general arrangement of the sample could be better explained graphically, e.g., how the plasmonic substrate is localized with respect to the drying front of the colloidal film and how/where exactly the drying droplet is illuminated and imaged. Right now, I am guessing (?) that it is at the edge of the droplet, but it really is not clear.

- Related to the point above, in Figure 1c, where are these images taken? At the edge of the droplet?

Response: Thank you for your comments; we will address both questions in a single response. The reviewer's observation is accurate – the images were indeed captured at the edge of the droplet.

ACTION TAKEN

To improve clarity, we have made changes to Figure 1 by including Figure 1(c), which provides a top-view illustration of the drying colloidal droplet, the plasmonic substrate (covering the entire substrate), and the position of the laser beam at the droplet's edge. Furthermore, we have revised the accompanying text to explicitly mention that the micrographs were taken at the droplet's edge. We believe that this visual representation will aid the reader in better comprehending the sequence of micrographs in Figure 1(d).

• The authors are describing the use of “Au spheres”. I would argue that such terminology is incorrect because metallic colloidal nanocrystals rarely are spherical but actually faceted crystals. Hence the word “sphere” is misleading.

Response: We agree with the reviewer, the term "sphere" is not accurate.

ACTION TAKEN

Consequently, we have amended the text, and we now refer to these materials as "dewetted Au nanoparticles" as has been previously employed in the literature (*ACS Appl. Nano Mater.* 2019, 2, 5, 3238–3245).

• When the Au “spheres” and bipyramids are introduced, they are not characterized in terms of their plasmonic properties and it is not motivated why two different types are used. This becomes clear only much later in the text when the different LPSR wavelengths are mentioned and there is referral to figures in SI. It would help the reader to understand this work if, at first mention, the different plasmonic properties of the two types of Au NPs used would be characterized (e.g. by an extinction spectrum of the colloidal solution) and shown, together with a short explanation of what can be achieved by having these two types of particles in terms of light manipulation/interaction.

Response: We appreciate the reviewer's suggestion and agree that the justification for choosing these two plasmonic systems should be provided earlier in the text.

ACTION TAKEN

To address this, we have incorporated an explanation at the end of the introduction with the following sentences. *"We engineer various colloidal plasmonic systems to spatially and temporally program the heating zones, acting as "virtual defects". More specifically, we investigated two systems: gold dewetted nanoparticles and gold bipyramids (Au BPs), which convert light into heat in the visible or near-infrared range, respectively. Moreover, we chose these two plasmonic systems to explore two different strategies: integrating the photothermal heaters on the substrate or within the colloidal solution."*

With these sentences, we aid the reader by anticipating the optical properties and the reasons for using the two plasmonic systems. However, we also believe that, for the sake of clarity and to avoid back-and-forth reading, the

detailed optical characterization of each plasmonic system is better integrated into the specific discussion of each experiment.

- In the context of Figure 2g, it is demonstrated that changing the arrangement and size of the plasmonic dots can be used to tune the crack pattern. Then it is also mentioned that the same effect can be obtained using colloidal NPs of two different types, for which laser wavelengths matching their respective LSPRs are used. Here, it would be relevant to showcase also control experiments where the respective samples are illuminated also off-resonance, i.e. the “spheres” with 852 nm and the bipyramids with 532 nm. An even more elegant and convincing experiment that I highly recommend is actually execute a scan across multiple wavelength below, at, and above the LSPR of each type of particle, to really reveal the importance of LSPR excitation.

Response: We agree with reviewer regarding scanning multiple wavelengths, which would indeed be very interesting and elegant. However, this presents significant technical challenges, as it would require the use of numerous (powerful and costly) lasers at different wavelengths that are currently beyond our reach. Nevertheless, in response to the reviewer's request, we conducted experiments where we excited the plasmonic particles with laser wavelengths that do not match their respective main Localized Surface Plasmon Resonance (LSPR). It's essential to note that **when exciting plasmonic particles off-resonance, we expect a reduced or negligible absorption of photons and conversion into heat.**

We characterized the temperature increase using an IR camera and analyzed crack deviation through image analysis. In the case of the dewetted particles irradiated with an 852nm laser, the IR camera showed no temperature increase, confirming that off-resonance irradiation does not induce photothermal heating. The Au bipyramids, irradiated with the different lasers, displays a wavelength depend crack deviation as shown in the newly added Figure S18.

Figure (a) illustrates the absorbance spectrum of colloidal Au bipyramids, which exhibit two absorption peaks. The spectrum displays a prominent longitudinal absorption peak at around 850 nm and a weaker transverse absorption peak at approximately 515 nm.

To assess the photothermal heating effect, we irradiated the colloidal system with lasers at different wavelengths (532 and 852 nm) while maintaining the same power of 3.39 W/cm². As shown in Figure (b), irradiation at 852 nm led to the maximum heating, reaching temperatures exceeding 100°C. In contrast, irradiation at 532 nm was less efficient, resulting in a temperature of approximately 40°C. This reduced efficiency can be attributed to the weaker transverse absorption peak and the mismatch with the resonance wavelength of the transverse absorption peak (515 vs 532 nm).

For reference, we compared these temperature values with the temperature of a system without Au bipyramids, which did not exhibit significant heating irradiated at 852 nm. When applied to the light-induced dip-coating experiment for deviating the cracks, 532nm irradiation results in a lower deviation compared to the 852nm, as shown in Figure (c). In summary, we confirm that exciting plasmonic particles off-resonance reduces photon absorption, heat conversion, and, consequently, crack deviation.

ACTION TAKEN

The Figure is now added in SI as Figure S18 and the discussion is integrated in the manuscript in page 13.

- In Figures 2 (h) and (i) the small bipyramid symbols are to illustrate different orientations according to the caption but they are oriented in the same way in the two panels?

Response: We appreciate the reviewer for pointing this out. We now realize that the caption and description may have been misleading. The small bipyramid symbols are not intended to represent pattern orientation; they are simply used to distinguish samples made of dewetted Au nanoparticles from those made of Au BPs.

ACTION TAKEN

To improve clarity, we have taken the following actions:

1) We have slightly modified Figure 2(b) to make the small symbols more visible as shown in the following inset:

2) We have clarified in the caption that the small symbols serve as a graphical guide to distinguish between samples made of Au dewetted nanoparticles and bipyramids, as follows: "*The small symbols will be used as a graphical guide to distinguish the sample made of Au dewetted nanoparticles or bipyramids.*"

3) Referring to Figure 2(h) and (i), in the caption, we have removed the term 'orientation,' which was causing confusion, and replaced it with "different hexagonal patterns".

• The experiments depicted in Figure 3 show how the crack formation in the drying colloidal film is impacted by laser illumination at the LSPR wavelength of Au bipyramids dispersed in the colloidal suspension. While the effect is clear, what I am lacking is an identical control experiment without Au nanoparticles in solution.

Response: As suggested we performed the identical control experiment by irradiating without Au BPs. As shown in the micrograph no crack deviation is observed without AuBPs. This is also in agreement with Figure S18(b) already discussed in a previous reply. We found that no heating was observed by irradiating solutions without AuBPs.

ACTION TAKEN

The micrograph is added in SI and the following sentence is added in page 14 "*As a consequence, without AuBPs, no deviation of cracks is observed (Figure S19).*"

• Figure 3 (g) is explained as cross-sectional "micrograph". However, it is not explained how it was obtained. Furthermore, in the caption "hyperspectral analysis" is mentioned. What is meant by this? In which way is this micrograph "hyperspectral"?

Response: these aspects need clarification.

Figure 3(g) is indeed a SEM micrograph displaying a cross-section view. This was achieved by cleaving the sample in the vicinity of the deviated cracks and observing it perpendicularly to the substrate plane using SEM.

ACTION TAKEN

To improve the clarity we:

changed the caption as follows "*SEM micrograph displaying a cross-section view of a deviated cracked zone.*" added further details in the Materials and Methods part when describing the SEM characterization: "*The SEM cross-section micrographs were obtained by cleaving the sample in the vicinity of the deviated cracks and observing it perpendicularly to the substrate plane using SEM*"

On the hyperspectral analysis: we indicated in the initial version that "the obtained hyperspectral image provides the scattering spectra of each pixel, enabling the monitoring of the optical evolution of the system^{40,41}" This can be seen as a 3D images in which the third dimension represents the wavelength (scattering in this case).

ACTION TAKEN

To further clarify this point, we modify the caption of Figure 3 as follows: "*Hyperspectral analysis of a single crack formation during the drying of a colloidal plasmonic droplet results in an image where each pixel provides a scattering spectrum, enabling spatial mapping of the plasmonic scattering.*"

• Figure 4 d, e, f in particular (but in many other figures as well) font sizes used are very small. Please improve overall readability of figures in this respect

Response: thank you for the suggestion. The font size of Figure 4 and also of the other Figures has been increased.

• The explanations given in Figure S19 are very important and I therefore suggest to actually include this in the main text.

Response: as suggested by the reviewer, we included the explanation about how to "anticipate" the path of cracks in the main text on page 16 (we will further discuss this point in the next reply).

• On line 286 (as well as in in SI Figure 20) the authors mention "predicted" crack patterns. How are they predicted? Is any kind of model used? This is unclear and nowhere explained?!

Response: Thank you for your comment; we will address this question in close relationship with the previous one. We agree, the term "predicted" can be confusing because we don't use a theoretical model. Through the analysis reported in 19 (now Figure S21), our goal was to we introduce a simple graphical approach to design the "expected" final crack pattern. As reported in SI and now in the text, the process involves several steps. The first step involves the design of deviated cracks as a function of their distance from the light spot. The extent of deviation for each crack is determined by the experimental values of ΔA and α , as previously identified. This results in the formation of an array of deviated cracks (Figure S21d). As discussed earlier, when two adjacent cracks come too close, "self-arresting" occurs. To account for this, the second step involves arresting the cracks when the convergence ratio reaches 0.74. The outcome of this simple process is referred now referred "expected" crack pattern (instead of predicted). This can be compared with the "experimental" crack pattern obtained after the actual experiment.

ACTION TAKEN:

1) the following sentences have been added to explain how to "anticipate" the crack patterns: "*From a given light pattern, we introduce a simple graphical approach to design the "expected" final crack pattern. Briefly, the process involves several steps. The first step involves the design of deviated cracks as a function of their distance from the light spot. The extent of deviation for each crack is determined by the experimental values of ΔA and α , as previously identified. This results in the formation of an array of deviated cracks (Figure S21d). As discussed earlier, when two adjacent cracks come too close, "self-arresting" occurs. To account for this, the second step involves arresting the cracks when the convergence ratio reaches 0.74, as depicted in Figure S21(f). The outcome of this simple process is referred to as the "expected" crack pattern and can be compared with the "experimental" crack pattern obtained after the actual experiment.*"

2) we changed the terms "predicted with "expected" in the text and in Figure 5 by making clear the difference between expected and experimental in Figure 5.

• On line 295 the authors mention "transparent" electrodes. It is unclear to me in which way this method could be used to make transparent electrodes when they are made from Au lines (which indeed interact with visible light?). Furthermore, what would be the advantage over making such electrodes simply with e.g. photolithography?

Response: Thank you for your comment; this point needs further clarification. As demonstrated in Figure 5, cracked films can serve as sacrificial masks to create metallic 'lines.' These metallic meshes, when applied to transparent substrates like glass or plastic, enhance the electrical conductivity of the surface without compromising its transparency. Previous examples of utilizing cracked films for fabricating transparent electrodes can be found in references 12 and 17, among others. In contrast to conventional photolithography, crack patterning enables the fabrication of smaller, sub-micrometric-wide metallic lines without the need for sophisticated equipment. As pointed out by the reviewer, Au lines reflect visible light, making the reduction in their width a crucial feature to maintain the transparency of the electrode.

ACTION TAKEN: Since this is not the core message of our article (but just a possible perspective), we decided to slightly modified the text by referring to the existing literature as follows " *...opening up intriguing possibilities for transparent electrodes as reported previously^{12, 17}.*"

REVIEWERS' COMMENTS

Reviewer #1 (Remarks to the Author):

The authors have addressed my revision requests in a convincing fashion and the manuscript can be published in its present form.

Reviewer #2 (Remarks to the Author):

After referee's suggestions (mine and others) most of the concerns suggested have been fulfilled by the authors. I can now recommend this revised version to be published in nature communications.

Reviewer #3 (Remarks to the Author):

I would like to thank the authors for a thorough and clear revision, in which all my comments have been carefully addressed. I can now fully recommend publication of this work in nature Communications.

REVIEWERS' COMMENTS

Reviewer #1 (Remarks to the Author):

The authors have addressed my revision requests in a convincing fashion and the manuscript can be published in its present form.

Reviewer #2 (Remarks to the Author):

After referee's suggestions (mine and others) most of the concerns suggested have been fulfilled by the authors. I can now recommend this revised version to be published in nature communications.

Reviewer #3 (Remarks to the Author):

I would like to thank the authors for a thorough and clear revision, in which all my comments have been carefully addressed. I can now fully recommend publication of this work in nature Communications.

REPLY: Once again, we would like to express our gratitude to the reviewers for their time and valuable suggestions.